

# Reconstructing the gluon

**Anton K. Cyrol[1], Jan M. Pawlowski[1,2], Alexander Rothkopf[1,3] and Nicolas Wink[1]**

**1** Institute for Theoretical Physics, Universität Heidelberg,
Philosophenweg 12, D-69120 Germany
**2** ExtreMe Matter Institute EMMI, GSI, Planckstr. 1, D-64291 Darmstadt, Germany
**3** Faculty of Science and Technology, University of Stavanger, NO-4036 Stavanger, Norway

## Abstract

We reconstruct the gluon spectral function in Landau gauge QCD from numerical data for the gluon propagator. The reconstruction relies on two novel ingredients: Firstly we derive analytically the low frequency asymptotics of the spectral function. Secondly we construct a functional basis from a careful consideration of the analytic properties of the gluon propagator in Landau gauge. This allows us to reliably capture the non-perturbative regime of the gluon spectrum. We also compare different reconstruction methods and discuss the respective systematic errors.

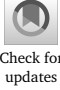
# 1 Introduction

Real-time correlation functions play a pivotal rôle for the theoretical understanding of heavy-ion collisions and the hadron spectrum. Their direct numerical computation in the strongly correlated regime of QCD is hampered by the fact that non-perturbative real-time methods are still in their infancies. Even though impressive results have been obtained in models and even in Yang-Mills theory, still more work is required on the way towards a full real-time approach to QCD.

In turn, non-perturbative Euclidean first principles methods such as Euclidean functional approaches and lattice simulations have been extensively used to obtain numerical results for QCD correlation functions. When analytically continuing these to the real-time regime or equivalently reconstructing their spectral function by means of solving an inverse integral transformation, one encounters large systematic uncertainties as in the case of single particle spectral functions [1–8] or the energy momentum tensor (EMT) [9–14], to name another pertinent correlator. Typically, the uncertainty even grows larger at small frequency. This is particularly harmful for the computation of transport coefficients which are related to the vanishing frequency limit of correlation functions of the EMT.

This problem of a large systematic uncertainty at low frequencies can be partially circumvented by a diagrammatic representation of the correlation functions of the EMT in terms of loops of real-time correlation functions of quarks and gluons, as discussed in [1, 4]. There the spectral function of the EMT has been computed from the single-particle spectral function of the gluon. The gluon spectral function itself, in the above mentioned paper has been reconstructed via a Bayesian spectral reconstruction method [15], which is a variant of the Maximum Entropy Method (MEM) [16, 17]. This approach also extends to general real-time computations of correlation functions on the basis of real-time single particle spectral functions of quarks and gluons.

A further reduction of the systematic error comes from prior information about the properties of the single particle spectral functions used as input. Often, such prior information is available for the high frequency asymptotics of the spectral function. This is the off-shell limit

with the Minkowski four momentum $p^2 \to \infty$.

In the present work we argue that the low frequency asymptotics is determined by the infrared (IR) limit in the Euclidean domain using only rather general assumptions. This leaves us with a well-constrained spectral function, which allows for a qualitatively enhanced spectral reconstruction. We apply this argument to the spectral reconstruction of the single particle gluon spectral function. The results presented here provide the starting point for the computation of transport coefficients in the spirit of the work presented in [1, 4], as well as direct real-time computation of thermodynamical properties and the QCD hadron spectrum.

This paper is organized as follows: In Section 2 we derive a general relation between the low frequency behavior of bosonic spectral functions and the infrared (IR) behavior of the corresponding Euclidean correlator. In Section 3 we summarize known properties of the gluon spectral function, its normalization and its asymptotics in the ultraviolet regime (UV). Then we turn to the low frequency behavior of the gluon spectral function in Section 4. Both the analytic structure of the scaling scenario and several realizations of the decoupling scenario are discussed. In Section 5, we reconstruct the gluon spectral function with a novel reconstruction method from numerical data from [18]. We conclude in Section 6 and provide a comparison of different reconstruction approaches in Appendix A.

## 2 Low frequency asymptotics of spectral functions

In general, a spectral function can be obtained from analytic continuation of its Euclidean propagator or from the inverse integral transformation via the Källén–Lehmann spectral representation. In this section we first introduce some basic definitions and then derive a novel general relation, (6), between the low frequency behavior of the spectral function and the infrared behavior of the Euclidean propagator. The relation is illustrated at a Breit-Wigner example before it is applied to the gluon spectral function in Section 4.

Throughout this section we assume that the propagator admits the Källén–Lehmann spectral representation

$$G(p_0) = \int_0^\infty \frac{d\lambda}{\pi} \frac{\lambda \rho(\lambda)}{\lambda^2 + p_0^2}. \tag{1}$$

The existence of a spectral representation has strong consequences for the analytic structure of the corresponding propagator, i.e. all non-analyticities are constrained to the $\mathrm{Re}\, p_0 = 0$ line. More details can be found in Appendix A. In (1) and in the following we have suppressed the momentum-dependence $\mathbf{p}$ of the spectral function and the propagator. Note that all arguments about $p_0 = 0$ apply equally to $p^2 = 0$ at vanishing temperature. In (1) the restriction to positive frequencies in the integral follows from the antisymmetry of the spectral function

$$\rho(\omega) = -\rho(-\omega). \tag{2}$$

Equivalently, the spectral function can be obtained from the Euclidean propagator by means of analytic continuation

$$\rho(\omega) = 2\,\mathrm{Im}\, G(-i(\omega + i0^+)), \tag{3}$$

i.e. from the discontinuity of the propagator. The low frequency behavior can directly be derived from (1), which is done in the following. We take a derivative of the spectral representation (1) with respect to the Euclidean frequency

$$\partial_{p_0} G(p_0) = -\int_{-\infty}^\infty \frac{d\lambda}{\pi} \lambda\, p_0 \frac{\rho(\lambda)}{(\lambda^2 + p_0^2)^2}. \tag{4}$$

We now take the limit $p_0 \to 0$ in (4) in order to access the low frequency behavior. With the derivative of the Poisson kernel representation of the delta function

$$\delta(x) = \lim_{\varepsilon \to 0} \frac{1}{\pi} \frac{\varepsilon}{\varepsilon^2 + x^2}, \tag{5}$$

one obtains the simple relation

$$\lim_{p_0 \to 0^+} \partial_{p_0} G(p_0) = -\frac{1}{2} \lim_{\omega \to 0^+} \partial_\omega \rho(\omega). \tag{6}$$

Equation (6) encodes the asymptotic behavior of the spectral function for small frequencies.

The low frequency behavior of spectral functions may also have an additional peculiarity at finite temperature, the transport peak. In case it is present, or in general at finite temperature, the limits of vanishing frequency $\omega \to 0$ and momenta $|\mathbf{p}| \to 0$ do not commute anymore, for a more careful discussion on this issue see e.g. [19]. Nevertheless, the analysis performed in this section holds, as all equations are viewed at fixed a $\mathbf{p}$.

As an example for the low frequency asymptotics we take a single pair of complex conjugated poles on the second Riemann sheet, i.e. a Breit-Wigner. The propagator is parametrized by

$$G^{(\text{BW})}(p_0) = \frac{A}{(p_0 + \Gamma)^2 + M^2}, \tag{7}$$

where $A$ is a suitably chosen normalization, $\Gamma$ is proportional to the width and $M$ is the mass. Expanding the derivative of (7) yields

$$\partial_{p_0} G^{(\text{BW})}(p_0) = -\frac{2A\Gamma}{(M^2 + \Gamma^2)^2} + \mathcal{O}(p_0). \tag{8}$$

Using (6), we obtain the small frequency behavior of the spectral function

$$\rho^{(\text{BW})}(\omega) = \frac{4A\Gamma\omega}{(M^2 + \Gamma^2)^2} + \mathcal{O}(\omega^2), \tag{9}$$

which is exactly the low frequency behavior of the full spectral function,

$$\rho^{(\text{BW})}(\omega) = \frac{4A\Gamma\omega}{4\Gamma^2\omega^2 + (M^2 + \Gamma^2 - \omega^2)^2}. \tag{10}$$

We emphasize that this derivation is based on the assumption of sufficient smoothness of the spectral function at low frequencies. A more careful derivation of (6), discussing the assumptions and other subtleties, such as modified spectral representations, is provided in Appendix A.

## 3 Known analytic properties of the gluon spectral function

Throughout this work we assume the existence of a spectral representation for the gluon propagator. This entails that the Euclidean gluon propagator $G_A(p)$ with Euclidean momenta $p = (p_0, \vec{p})$ can be written in terms of a gluon spectral function $\rho_A(\lambda, \vec{p})$ as

$$G_A(p_0) = \int_0^\infty \frac{d\lambda}{\pi} \frac{\rho_A(\lambda)}{\lambda^2 + p_0^2} + \sum_{j \in \text{poles}} \frac{R_j}{p_0^2 + M_j^2}, \tag{11}$$

analogously to (1), where $M_j \in \mathbb{C}$ are the position of poles with $\mathrm{Im}\, M_j \neq 0$ and the $R_j$ the corresponding residues. As there we have suppressed the spatial momentum dependence in (11). In (11) we also allowed for additional poles for the sake of maximal generality. In most cases these poles will be discarded.

The existence of a spectral representation of the gluon is implicitly underlying various relations and properties used in covariant approaches to QCD. In the present context this is most apparent -but by now means restricted to- for the pinch technique, see e.g. [20]. Further works implicitly using gluon spectral representation can e.g. be found in [21,22].

While the low frequency properties discussed in the preceding section apply to any bosonic spectral function, we now turn to the gluon spectral function as a specific example. Their normalization and high frequency asymptotics are well-known properties of QCD, which we briefly discuss next. They are exploited in the reconstruction of gluon spectral functions from Euclidean data in Section 5.

The normalization of the gluon spectral function in Landau gauge follows from the Oehme-Zimmermann superconvergence relation [23,24],

$$\int_0^\infty \mathrm{d}\lambda\, \lambda\, \rho_{\mathrm{A}}(\lambda) = 0, \tag{12}$$

(12) entails that $\rho_{\mathrm{A}}(\lambda)$ has positive *and* negative values in contrast to spectral functions of physical (asymptotic) states, e.g. hadronic spectral functions. Then, the total spectral weight is finite and is typically normalized to one. Its conversation is related to unitarity. When reconstructing the gluon spectral function, (12) serves as a highly non-trivial consistency check. Alternatively one may simply enforce it within the reconstruction method as part of the prior information.

Let us further consider the high frequency behavior of the spectral function. The asymptotic off-shell propagator can be determined in perturbation theory for arbitrary $p_0 \in \mathbb{C}$ [25]. In pure glue theory the only scale is the dynamical QCD scale $\Lambda_{\mathrm{QCD}}$. Accordingly, its asymptotic regimes are define with the limits of the dimensional momenta and frequencies,

$$\hat{p}^2 = \frac{p^2}{\Lambda_{\mathrm{QCD}}^2}, \qquad \hat{\omega} = \frac{\omega}{\Lambda_{\mathrm{QCD}}}. \tag{13}$$

With the dimensionless momentum $\hat{p}$ the dimensionless propagator $\hat{G}_{\mathrm{A}} = \Lambda_{\mathrm{QCD}}^2 G_{\mathrm{A}}$ reads

$$\lim_{\hat{p}^2 \to \infty} \hat{G}_{\mathrm{A}}^{(\mathrm{UV})}(p) = \frac{Z_{\mathrm{UV}}}{\hat{p}^2 \log(\hat{p}^2)^\gamma} + \text{sub-leading}, \tag{14}$$

where $Z_{\mathrm{UV}}$ is an overall dimensionless normalization and

$$\gamma = \frac{13}{22}, \tag{15}$$

is the one-loop anomalous dimension of the gluon. Using (3) one can easily obtain the asymptotic behavior of the dimensionless spectral function $\hat{\rho}_{\mathrm{A}} = \Lambda_{\mathrm{A}}^2 \rho_{\mathrm{A}}$ with

$$\lim_{\omega \to \infty} \Lambda_{\mathrm{QCD}}^2 \rho_{\mathrm{A}}^{(\mathrm{UV})}(\omega) = -\frac{Z_{\mathrm{UV}}}{\hat{\omega}^2 \log(\hat{\omega}^2)^{1+\gamma}} + \text{sub-leading}. \tag{16}$$

One key aspect is the leading order negativity of the spectral function at high frequencies. As a direct consequence, the gluon admits positivity violation in Landau gauge and cannot be an asymptotic state of the theory. As a consequence, the spectral function cannot be interpreted in the usual probabilistic sense anymore. Further details can be found in e.g. [25–27].

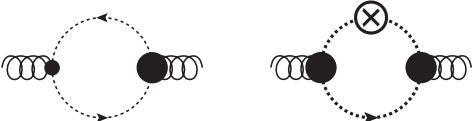

Figure 1: Ghost loops in the gluon propagator DSE (left), [28] and FRG (right), [18]. Internal dashed (wiggly) lines represent fully dressed ghost (gluon) propagators. Thin (thick) blobs depict undressed (dressed) vertices. The crossed circle in the FRG diagram denotes the regulator insertion. The massless ghost propagator causes these diagrams to yield logarithms as given in (26). The dressing of ghost-gluon vertex cannot change this qualitative behavior since it is constant in the infrared, see Figure 2.

## 4 Low frequency properties of the gluon spectral function

In the present section we evaluate the novel frequency relation of (6) for the gluon spectral function. We show that for the scaling solution in the Landau gauge, derived from the Kugo-Ojima criterion [29], the low frequency asymptotics is negative. For the decoupling solution found on the lattice and in various analytic approaches the situation is less clear. However, for the expansion schemes used in the literature we also find negative spectral functions.

The derivation of (6) has been quite general and holds for a large class of operators. The application of (6) only requires the knowledge of the asymptotic infrared (IR) behavior of the theory at hand. Despite the tremendous progress in understanding the IR sector of Yang Mills theory and QCD, we still lack a comprehensive picture. Various approaches have been put forward to predict the analytic IR behavior of the Euclidean gluon propagator, which we use to determine the small frequency behavior of the gluon spectral function in the following. In general the Landau gauge gluon propagator $\hat{G}_A = \Lambda^2_{\text{QCD}} G_A$ in the deep IR can be parametrized by

$$\hat{G}_A(p_0) = Z_{\text{IR}} x^{-1+2\kappa}, \tag{17}$$

with the scaling coefficient $\kappa$ and an overall dimensionless IR normalization $Z_{\text{IR}}$ and

$$x = \hat{p}^2 + \gamma_G \left( m^2_{\text{gap}} + z_G \, \hat{p}^2 \ln \hat{p}^2 \right), \tag{18}$$

with $z_G > 0$. The remainder of this section concerns the structure of (17), additionally all equations are understood in the deep IR limit, i.e. $\hat{p}^2, |\hat{\omega}| \ll 1$.

The parameter $\gamma_G \in [0, 1]$ is related to the Gribov ambiguity, together with an appropriate definition of $m^2_{\text{gap}}$. The lower limit $\gamma_G \to 0$ recovers the *scaling* solution, while the upper limit $\gamma_G \to 1$ can be considered as implementing the maximal breaking of global BRST symmetry. In the following we call the set of solutions with $\gamma_G > 0$ *decoupling*. Despite their differences in terms of scaling both solutions have in common that their $p^2$-derivative diverges in the infrared,

$$\lim_{p^2 \to 0} |\partial_{p^2} G_A(p^2)| \to \infty. \tag{19}$$

For the scaling solution it follows with $0 < \kappa < 1$ and $\kappa \neq 1/2$. For the decoupling solution the divergence originates in the logarithm $\ln p^2$. Moreover, from (6) it follows that it is precisely the sign of the $p^2$-derivative in (19) which determines the sign of the spectral function for low frequencies,

$$\text{sgn}\left[\lim_{\omega \to 0} \rho_A(\omega)\right] = -\text{sgn}\left[\lim_{p^2 \to 0} \partial_{p^2} G_A(p^2)\right], \tag{20}$$

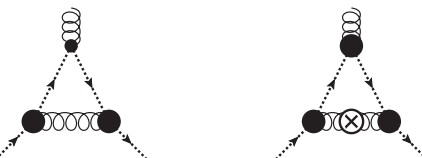

Figure 2: Typical diagrams that contribute to the ghost-gluon vertex DSE (left), [40] and FRG (right), [18]. The presence of the gapped gluon propagator ensures that the ghost-gluon vertex is constant in the infrared. It can be shown that this behavior is present at every finite truncation level.

where we used that the sign of the spectral function and its derivative are identical at low frequencies. This follows from the expansion of the spectral function around zero

$$\rho_A(\omega) = \omega \, \partial_\omega \rho_A(\omega) \tag{21}$$

for positive frequencies. In (21) the vanishing of the zeroth order, i.e. $\rho_A(0) = 0$, is one of our basic assumptions, c.f. the discussion in Appendix A. Equation (20) entails that the backbending of the propagator leads to a negative spectral function at low frequencies. Note that a backbending implies the existence of a gluon propagator maximum at a finite momentum which indicates positivity violation, see e.g. [26].

Apart from the low frequency behavior we are also interested in the analytic structure of the gluon propagator. The latter is relevant for an accurate determination of the quasi-particle peak we expect at frequencies related to the physics scale $\Lambda_{\text{QCD}}$: The analytic form of (17) is exact for the scaling solution, see e.g. [30–38], and the discussion in Section 4.1. For the decoupling solution (17) has to be seen as an ansatz. In particular, it is one that is motivated from an Euclidean perspective and it may introduce ambiguities regarding the analytic structure of the propagator in the complex plane. Different proposals for the analytic structure of the gluon propagator have already been made in [39], one of which is compatible with the scenario discussed here. We postpone the thorough discussion of the parametrization to Section 4.2 and Section 4.3.

## 4.1 Scaling solution

The scaling solution is obtained by setting $\gamma_G = 0$ in (17). The asymptotic behavior of the gluon propagator then reads

$$\hat{G}_A^{(\text{sca})}(p) = Z_{\text{IR}} (\hat{p}^2)^{-1+2\kappa} . \tag{22}$$

The scaling coefficient $\kappa$ is constrained by $1/2 < \kappa < 1$ and recent numerical calculations suggest $\kappa \approx 0.58$ [18] in Yang-Mills theory.

Combining (6) and (22) we obtain the following low frequency asymptotics of the gluon spectral function for the scaling solution

$$\hat{\rho}_A^{(\text{sca})}(\omega) = -2 Z_{\text{IR}} \operatorname{sgn}(\hat{\omega}) (\hat{\omega}^2)^{-1+2\kappa} . \tag{23}$$

Most notable is the negative sign, i.e. the spectral function is negative for small positive frequencies. The functional similarity between (22) and (23) is not very surprising since the scaling solution has a rather simple complex structure, a single branch cut at $\operatorname{Re} p_0 = 0$.

## 4.2 Decoupling Solution

In this section we discuss in detail the infrared behavior of the decoupling solution. In contradistinction to the scaling solution where the analytic structure follows directly from scaling

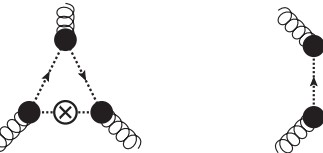

Figure 3: Ghost triangle (left) and ghost box (right) diagrams as they appear in the three- and four-gluon vertex flow equations [18]. As is well known, these ghost loops generate logarithmic divergences in the vertices. Similar diagrams contribute to the respective vertex DSEs, see e.g. [41–45].

in the Euclidean regime, in the decoupling case this necessitates to monitor the infrared leading logarithms. While the leading logarithms are fully accessible, a complete analysis requires to take into account the back-coupling of the quantum corrections in the functional equations.

To begin with, the leading behavior for the decoupling solution strictly speaking reads

$$\hat{G}_{\mathrm{A}}^{(\mathrm{dec})}(p_0) \sim \frac{1}{x}, \tag{24}$$

with $x$ given by (18) and $\hat{G}_{\mathrm{A}} = \Lambda_{\mathrm{QCD}}^2 G_{\mathrm{A}}$. (24) is the leading term of (17) in an expansion around $p_0 = 0$. The log-term in (18) arises naturally from the momentum integration of the ghost loop in the IR, see Figure 1. There we have depicted the ghost loop in both the functional renormalization group (FRG) equation for the gluon propagator, see e.g. [18] for more diagrammatic details, and the Dyson-Schwinger equations (DSE), see e.g. [26] for more diagrammatic details. Both depend on the ghost propagator and the ghost-gluon vertex. In the decoupling case the ghost propagator has a trivial infrared behavior proportional to $1/p^2$.

As a side remark we mention that the ghost propagator is not augmented with a leading order logarithmic IR running, even though this would not change the present analysis. The absence of a leading order logarithmic IR running in the ghost propagator can be shown along a similar line of arguments as done here for the gluon propagator.

### 4.2.1 Sources for infrared logarithms

We now proceed with the discussion of the IR behavior of the gluon propagator. The ghost-gluon vertex dressing tends towards a constant value with a small angular dependence for small momenta, while the ghost propagator dressing also tends towards a constant. The low momentum triviality of the ghost-gluon vertex is related to the non-renormalization theorem for the ghost-gluon vertex in the Landau gauge. It also can be seen from Figure 2 which features the gapped gluon propagator and hence is infrared suppressed. This property holds for all mixed ghost-gluon correlations. For a very detailed and extensive discussion in the context of a perturbative one-loop analysis of the Curci-Ferrari setup we refer to [46]. In summary the ghost loop depicted in Figure 1 gives rise to a $p^2 \ln p^2$ contribution with a negative prefactor in the IR. This is reflected by $z_G > 0$ in (18). The other diagrams contain the gapped gluon propagator or non-classical vertices. Consequently these diagrams cannot contributed to the logarithmic running at one-loop.

Beyond one-loop further contributions to the IR logarithm could originate from the logarithmic running of the vertices. This scenario was behind the discussion of the Higgs phase for large explicit gluon masses in [18]. These contributions would have the potential of switching the sign of $z_G$. Again such a running can only be triggered by ghost loops due to the gapping of the gluon. Hence, from an iterative point of view, they first can only occur for purely gluonic vertices triggered by the massless ghost loops contributing to these vertices. If created, they can propagate to all correlation functions via diagrams with at least one purely gluonic vertex.

For the propagator the three- and four-gluon vertices are relevant, for the respective diagrams see Figure 3. Indeed these vertices feature logarithmic terms at one loop, see e.g. [18,41–45].

The mere occurrence of logarithmic terms in the vertices is not sufficient for triggering an additional logarithmic running of the gluon propagator. Consider for example a logarithm of the form $\ln(p^2 + q^2)$, where $q$ is the loop momentum of a given diagram for the gluon propagator $G_A(p)$. Then the loop integration effectively removes this logarithm as the gluon in the diagram is gapped. Consequently only logarithmic terms of the form $(p_i)_\mu f_i(p_1, p_2) \ln p^2$ for the three gluon vertex, and $f(p_1, p_2, p_3) \ln p^2$ for the four-gluon vertex would trigger $p^2 \ln p^2$-terms in the propagator. Here $p$ is one of the momenta $p_1, ..., p_n$ with $p_n = -(p_1 + \cdots + p_{n-1})$ in an $n$-gluon vertex.

Even though the presence of such terms would be of great interest for the effective detection of a possible Higgs phase [18], a complete analysis is beyond the scope of the present work. Here we simply remark that the terms of the required form are singled out by the infrared limit of one momentum $p \in (p_1, ..., p_n)$ with

$$\lim_{\hat{p}^2 \to 0} |\partial_{p^2} \Gamma^{(n)}_{\text{gluonic}}(p_1, ..., p_n)| \to \infty \,, \tag{25}$$

for the three- and four gluon vertices, $n = 3, 4$, at fixed other momenta. For this limit one can concentrate on the propagators attaching the ghost-gluon vertex with the momentum $p$. In the above limit they only carry the loop momentum, but are multiplied by $q_{\mu_i}$ from the respective ghost-gluon vertices. Hence they diverge as $1/q^2$. The derivative w.r.t. $p^2$ triggers another $1/q^2$: Applied to th ghost propagator $G_c$ that carries the external momentum $p$, we are led to $\partial_{p^2} G_c(q + p)^2 \propto (1/q^2)^2$ in the limit $p \to 0$. In summary this leaves us with a logarithmic singularity due to $d^4q \, 1/q^4$. The other propagators in the diagrams still carry other external momenta and do not add to the singularity.

In summary, the kinematic analysis above hints at the existence of the logarithmic terms in the gluonic vertices that act as additional sources for the logarithmic running of the propagator. Note however, that a decisive answer requires an analysis that also takes into account the underlying gauge symmetry: first of all the Slavnov-Taylor identities (STIs) connect the different diagrams in the functional equations for the gluon propagator. Second, the STIs also restrict the vertex structures themselves and the prefactors of the logarithmic vertex corrections may even vanish for fully dressed vertices. It goes without saying that even for being indicative such an analysis requires at least a full two-loop analysis of the gluon propagator in the presence of a mass gap. In this context we mention a very careful complete and illuminating perturbative analysis at one-, two and three loops in [47–49] in QCD and [50, 51] in Curci-Ferrari-type models, and also references therein.

Accordingly, the logarithms produced always depend on sums of combinations of external momenta squared. This kinematic argument entails that vertex logarithms always depend on loop momenta and hence do not contribute to $z_G$. Note that this argument, upon iteration, holds for fully non-perturbative resummations as done within functional methods. We emphasize that evidently this proof necessitates both the logarithmic corrections of vertices as well as the logarithmic corrections that originates from the massless propagators in the loop. Hence, conclusive arguments should at best make systematic use of the full iterative structure of resummation schemes as done here, or exploit perturbation theory at two loop and beyond. The latter ensures in most cases that the perturbative structure mimics of the iterative structure of non-perturbative resummations.

### 4.2.2 Potential Higgs branch

For its relevance we come back to the Higgs-phase argument in [18], even though it is a bit outside the line of arguments here. The existence and properties of such a Higgs phase are

not only important for the Standard model but also for finite temperature QCD, where the temporal gauge field plays the rôle of a Higgs field. In [18] the dynamics of such a Higgs field was trivially mimicked by an explicit mass term of the gluon despite of its dynamical structure. The present analysis makes it apparent why such an argument falls short. In the presence of a Higgs mechanism one resorts to $R_\xi$-gauges that leads to massive ghosts in the Higgs phase with the ghost mass proportional to the expectation value of the Higgs. Within the present setup this has been discussed in [52]. There it has been also shown that this mechanism has an equivalent in the standard Landau gauge. In Landau gauge the Higgs-Kibble dinner is not apparent. Still, the effect of the massless ghosts is more than balanced by that of the Goldstone modes. In [52] it has been shown that this leads to a deconfining Polyakov loop potential in the Higgs phase. In the present context it entails that the Goldstone contributions to the gluon in Landau gauge are an additional source of the $p^2 \ln p^2$ running of the gluon propagator, that can turn the sign of $z_G$: this simply follows from the similarity of the Higgs-gluon vertex to that of the ghost-gluon vertex and a respective perturbative analysis. A more detailed analysis is far beyond the scope of the present work and deferred to future work.

### 4.3 Scenarios for analytic structures of the decoupling solution

Now we proceed with our main line of arguments. Even though sufficiently smooth, the non-trivial angular dependence and the sub-leading momentum-dependence will still almost certainly modify the complex structure. Nonetheless (18) still provides a very good parametrization in the infrared. Accordingly, in contradistinction to the scaling solution it is not possible for the decoupling solution to determine its analytic structure from the IR asymptotics. The difference between parameterizations cannot be resolved in currently available Euclidean data as the effects are sub-leading in the Euclidean IR domain. Nevertheless, the basic form and generation of terms is well motivated and an investigation of the IR behavior is still sensible for the case of the decoupling solutions. It allows us to classify two likely scenarios for the analytic structure of the decoupling type gluon propagator:

#### 4.3.1 Scenario I

We start with the parametrization given in (17) since it is the simplest one capturing the Euclidean behavior. Keeping a finite $\gamma_G$ in (17) this parametrization of the decoupling propagator can be reduced to

$$\hat{G}_A^{(\mathrm{dec})}(p) = \tilde{Z}_{\mathrm{IR}} \left( \tilde{m}_{\mathrm{gap}}^2 + \hat{p}^2 \ln \hat{p}^2 \right)^{-1}, \tag{26}$$

after absorbing $\gamma_g$ and $z_G$ by appropriate redefinitions of $Z_{\mathrm{IR}} \to \tilde{Z}_{\mathrm{IR}}$ and $m_{\mathrm{gap}}^2 \to \tilde{m}_{\mathrm{gap}}^2$. The parametrization (26) admits complex conjugated poles, which lead to a modification of the simple spectral representation (1). Allowing for additional poles, we make use of the extended spectral representation (11). This enables us to separate cut and pole contributions of (26), a detailed description of the analytic structure can be found in Appendix B. Specializing (26) to the contribution of the cut, i.e. the one contributing to $\rho_A^{(\mathrm{dec})}$ we obtain with (6)

$$\hat{\rho}_A^{(\mathrm{dec})}(\omega) = -\hat{Z}_{\mathrm{IR}} \, \mathrm{sgn}(\hat{\omega}) \frac{2\pi}{\hat{m}_{\mathrm{gap}}^4} \hat{\omega}^2 + \mathcal{O}(\hat{\omega}^4 \ln \hat{\omega}). \tag{27}$$

Again, most notable is the negative sign in front of (27), leading to a negative spectral function at low frequencies for the parametrization (26) of the decoupling solution.

### 4.3.2 Scenario II

As already mentioned above, the form (26) is not unique, and cannot be fixed by presently available data. Indeed, another admissible parametrization removes the additional poles in (26). Then the propagator exhibits a single cut. We keep the same leading order expansion in $p_0 = 0$, which renders all differences sub-leading in the Euclidean data in the IR. A possible parametrization with these properties is given by

$$\hat{G}^{(\mathrm{dec})}(p) = \tilde{Z}_{\mathrm{IR}} \tilde{m}_{\mathrm{gap}}^{-2} \left( 1 + \tilde{m}_{\mathrm{gap}}^{-2} \ln\Gamma(\hat{p}^2) \right). \tag{28}$$

Here, $\ln\Gamma$ is the logarithmic $\Gamma$ function (not the logarithm of the $\Gamma$ function). The logarithmic $\Gamma$ function has a branch cut for $\mathrm{Re}\, z = 0$ and is analytic everywhere else. It is defined by

$$\ln\Gamma(z) = \sum_{k=1}^{\infty} \left( \frac{z}{k} - \ln\left(1 + \frac{z}{k}\right) \right) - \gamma_{\mathrm{E}} z - \ln(z), \tag{29}$$

with the Euler-Mascheroni constant $\gamma_{\mathrm{E}}$. Both parameterizations lead to the same leading order term in the propagator,

$$\hat{G}^{(\mathrm{dec})}(p) = \tilde{Z}_{\mathrm{IR}} \tilde{m}_{\mathrm{gap}}^{-2} \left( 1 - \tilde{m}_{\mathrm{gap}}^{-2} \hat{p}^2 \ln(\hat{p}^2) \right) + \mathcal{O}(\hat{p}^4). \tag{30}$$

This implies the same low frequency behavior of the spectral function as in (27) and demonstrates explicitly the remaining freedom in parameterizing the decoupling solution while possibly modifying the corresponding spectral function. It is important to note however that the leading term in $\partial_{p^2} G$ is the one containing the logarithm, whose sign cannot be flipped and from which the non-positivity of the small frequency spectrum arises. More details regarding the propagator at zero can be found in Appendix A. We close this section with a word of caution: While a large class of parameterizations may yield the same spectral function, as it is the case here, this is by no means guaranteed.

## 4.4 Realizations of decoupling solutions

In the following subsections several approaches or models that feature decoupling type solutions are discussed. In most cases the gluon propagators results in the approaches are worked out in specific expansion schemes that allow us assigning one the above scenarios described in Section 4.3.1, Section 4.3.2 to it. Note that this does not necessarily entail the correct analytic structure of the gluon propagator in the given approach but certainly that of the given expansion order. Note also, that the systematics of generic expansion schemes in the analytic functional approaches suggests the persistence of the analytic structure if resummed vertices are taken into account. However, a detailed analysis is beyond the scope of the present work.

### 4.4.1 Lattice

Our discussion is based on the plethora of lattice results for decoupling gluon propagators at vanishing and finite temperature [53–57,57–66], for recent analytic fits to high precision data see, e.g., [67] and [68]. However, in our opinion the distinction between the different scenarios Section 4.3.1, Section 4.3.2 still requires a far higher precision. Accordingly, without additional information it is not possible to differentiate between any of the possible parameterizations of the decoupling scenario. Therefore statements about the analytic structure of the gluon propagator based on lattice data is currently not possible.

Several reconstructions based on lattice data have been performed in the past years. In [3] a reconstruction was presented using simulations results from low temperature quenched

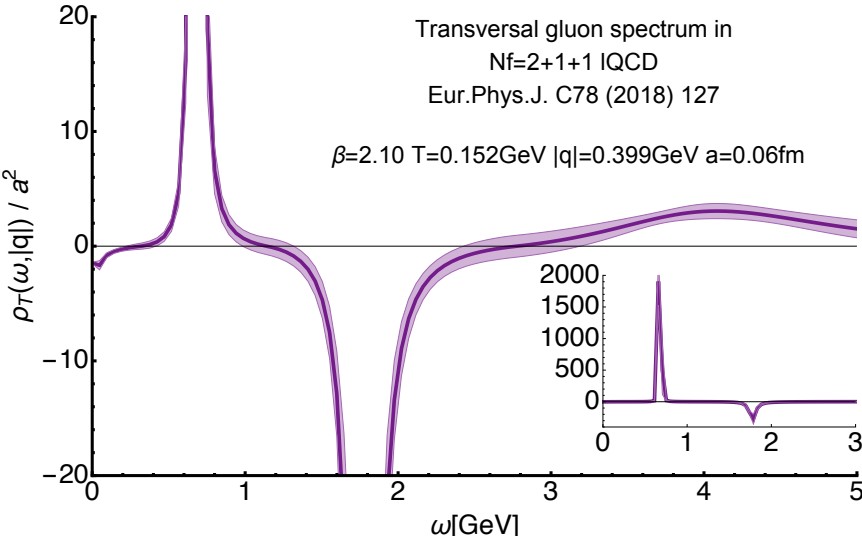

Figure 4: Example of a gluon spectral function in the confined phase $T = 152\,\mathrm{MeV} < T_C$ extracted from lattice QCD simulations by the tmft collaboration including $N_f = 2 + 1 + 1$ flavors of quarks. Note that while the higher lying negative feature at around 1.75 GeV is strongly pronounced, we also find indications for a residual negative contribution at small frequencies. The overshoot into positive values at higher frequencies originally thought of as a Bayesian artifact also emerges in our reconstruction presented below. Error bars include both statistical and systematic errors, for details see [6].

lattice QCD. The authors deployed the Tikhonov regularization to extract the spectral function and observed a negative contribution at low frequencies.

Finite temperature studies have also been carried out. A reconstruction including finite temperature gluon propagators in quenched QCD based on a modified Maximum Entropy Method (MEM) was presented in [1]. The results were mostly positive for small frequencies by construction, due to the modified MEM approach for non-positive spectra. This method has also been applied to decoupling FRG data, see Section 4.4.2, and the two reconstructions give similar results.

Another reconstruction based on a Bayesian approach has been performed in [6] using finite temperature lattice QCD data, featuring $N_f = 2 + 1 + 1$ quark flavors. The generalized Bayesian Reconstruction (gBR) approach [5] deployed in that study revealed that in the confined phase the gluon spectrum exhibits a small residual negative contribution at small frequencies, see Figure 4. Any sign of this negative structure disappeared at higher temperatures, however the systematic uncertainties in the study precluded a definite statement, whether that was a genuine finite temperature effect. We see the finding of a negative low frequency part as a strong indication that the Bayesian reconstruction method (gBR) in [6] recognizes the low frequency relation derived in this work.

In summary, the discussion of the low frequency limit of the gluon spectral function and of the analytic structure suggests to revisit the spectral reconstruction of the gluon spectral function based on improved analytic models that incorporate the logarithmic corrections of the gluon propagator. As the logarithmic terms might be difficult to extract directly even from the high precision lattice data, it calls for a combined lattice-functional methods approach: the logarithmic terms could be constrained by using combined propagator and vertices lattice data and lattice consistent results from functional methods. In the latter the logarithmic infrared terms can easily be extracted.

#### 4.4.2 DSE and FRG

Decoupling-type propagators have been computed in both DSE and in FRG calculations in good agreement with the corresponding lattice results, see e.g. [28,69,70] and [18,28], respectively. Within the DSEs a direct solution has been computed in the complex plane in [71], where a single branch cut along the $\mathrm{Re}\,p_0 = 0$ axis was found. The spectral function found there stays positive for very small frequencies. Hence the analytic structure has to violate implicitly our smoothness condition, which is very interesting and requires a more detailed investigation.

As mentioned already in the previous Section 4.4.1, decoupling FRG data as well as lattice data for the finite temperature gluon propagator have been used for a reconstruction of the gluon spectral function for temperatures $T \geq 100\,\mathrm{MeV}$ in [1]. Both, the reconstruction of the lattice data and that of the FRG data have been in very good agreement with each other.

Moreover, the results are in qualitative agreement with that of the direct DSE computation of [71]: The finite temperature data show a thermal broadening. The MEM-type method used in [1] run into accuracy problems for smaller temperatures. This is a typical sign of a sharp peak in the spectral function. A low temperature extrapolation of the thermal spectral functions gives rise to a sharper peak at $T = 0$, but no quantitative statement was possible due to the missing small temperature accuracy. Note also, that the reconstruction method used in [1] leads to a positive low frequency tail almost by construction. Apart from this disagreement the results there are also in qualitative agreement with the reconstruction of the scaling spectral function presented in Section 5.4 shown in Figure 6.

In summary, as already mentioned at the end of Section 4.4.1, the situation calls for a combined lattice-functional methods approach in order to minimize the systematic error of the reconstruction.

#### 4.4.3 Gribov-Zwanziger approach

The complex structure arising in the Gribov-Zwanziger approach has been discussed in [72] at the example of a toy model with complex conjugated poles. The current state of the art comparison with lattice data [68] resorts to a tree-level propagator with a perturbative RG improvement that captures the ultraviolet running. It reads

$$G_A(p^2) = \frac{p^2 + M_1^2}{p^4 + M_2^2 p^2 + M_3^4} \left[ \ln \frac{p^2 + m_g(p^2)}{\Lambda_{\mathrm{QCD}}^2} \right]^\gamma, \tag{31}$$

with the one-loop anomalous dimension $\gamma = 13/22$ introduced in (15). The regularization mass $m_g(p^2)$ is finite in the IR for $p^2 \to 0$ and either decays or also stays finite in the UV for $p^2 \to \infty$. (31) is sufficient to capture the high frequency behavior as well as the non-perturbative gapping of the gluon. Its complex structure features the perturbative cut as well as complex conjugated poles. The spectral function that follows from the propagator (31) is subject to the infrared relation (6). Evidently, the sign of the spectral function depends on the combination of parameters chosen in (31), for the best fits provided in [68] it is negative.

A one-loop analysis of the GZ approach reveals a logarithmic IR momentum scaling that originates in the gauge-fixing contributions similar to the ghost contribution in the Landau gauge. (31) lacks this logarithmic IR running that leads to the negative sign of the spectral function for low frequencies. As it is not built in naturally in (31) it suggests to simply restrict the range of allowed parameters by

$$\lim_{p^2 \to 0} \partial_{p^2} G(p^2) > 0, \tag{32}$$

which mimics the divergent limit (19) insofar that it reproduces (20), and hence the correct sign of the spectral function at low frequencies. Alternatively the propagator model (31) can be amended by an cut. In either case this enhances the predictive power of the reconstruction.

### 4.4.4 Curci-Ferrari model

The Curci-Ferrari model [73] is a massive version of Yang-Mills theory. As such it features an additional relevant coupling, the gluon mass, and reduces to Yang-Mills theory in the - appropriate- massless limit. In recent years, the model has seen revived interest in the context of modeling the non-perturbative mass gap of QCD with a respective choice of the Curci-Ferrari mass parameter. Then, a perturbative treatment of fluctuations may be possible. This reasoning has been introduced in [74–76] where QCD correlation functions have been modeled using perturbation theory, for a recent work see [46]. In the present context this is particularly interesting, as it also allows analytically accessing the kinematic arguments given in Section 4.2.

The one-loop contribution to the gluon propagator has been calculated and discussed in [46, 74]. It features an asymptotic IR behavior of the form (26), its infrared properties and the relation to positivity violation have been discussed extensively in [46]. We are led to a negative low frequency spectral function of the form (27). Higher loop considerations may change the global cut form as discussed in Section 4.2, but are not relevant for the question of the low frequency behavior.

A very detailed analysis of the complex structure of the Curci-Ferrari model is also found in [77–79]. In particular in [79] the gluon propagator in the CF-model is worked out at one loop, leading to a low frequency spectral function with (27).

In summary, the detailed one loop analysis in [46, 79] shows the low frequency properties of the spectral function derived here, (6). Concerning the global complex structure a two-loop analysis in the CF-framework with respect to its complex structure would be very interesting as it features both one-loop dressed propagators and vertices. In this context we refer the reader to [50, 51] where Curci-Ferrari-type models have been studied up to three loop. A respective analysis should also provide valuable additional information for the reconstruction of Landau gauge spectral function in general.

## 5 Extracting the spectral function from the Euclidean propagator

The aim of this section is to reconstruct the gluon spectral function from numerical data of the gluon propagator obtained in the scaling scenario [18]. The final result for the spectral functions is shown in the left panel of Figure 6. We would like to emphasize that it exhibits all the analytic properties discussed above up to numerical uncertainties.

If used to compute the propagator via (1) it reproduces the original input with a precision of $\sim 2\%$, as shown in the right panel of Figure 6.

To arrive at the spectral function we use a novel approach based on an explicitly constructed set of basis functions that is carefully derived from the analytic properties of two-point correlation functions, see Figure 5 for an illustration. First, we discuss the underlying analytic structure. Next, we introduce the explicit form of the basis and finally describe how it is applied to extract the gluon spectral function.

### 5.1 Analytic structure of the retarded propagator

The picture we have used in the preceding sections assumes a specific analytic structure for the gluon propagator, i.e. that it contains a single branch cut at $\operatorname{Re} p_0 = 0$ (for more details see Appendix A). Therefore, we have two analytic patches in the complex plane, the retarded propagator for $\operatorname{Re} p_0 > 0$ and the advanced one for $\operatorname{Re} p_0 < 0$. They are related by the well-known relation

$$G^{(\mathrm{ret})}(-\mathrm{i}(\omega + \mathrm{i}\varepsilon)) = \left[ G^{(\mathrm{adv})}(-\mathrm{i}(\omega - \mathrm{i}\varepsilon)) \right]^* . \tag{33}$$

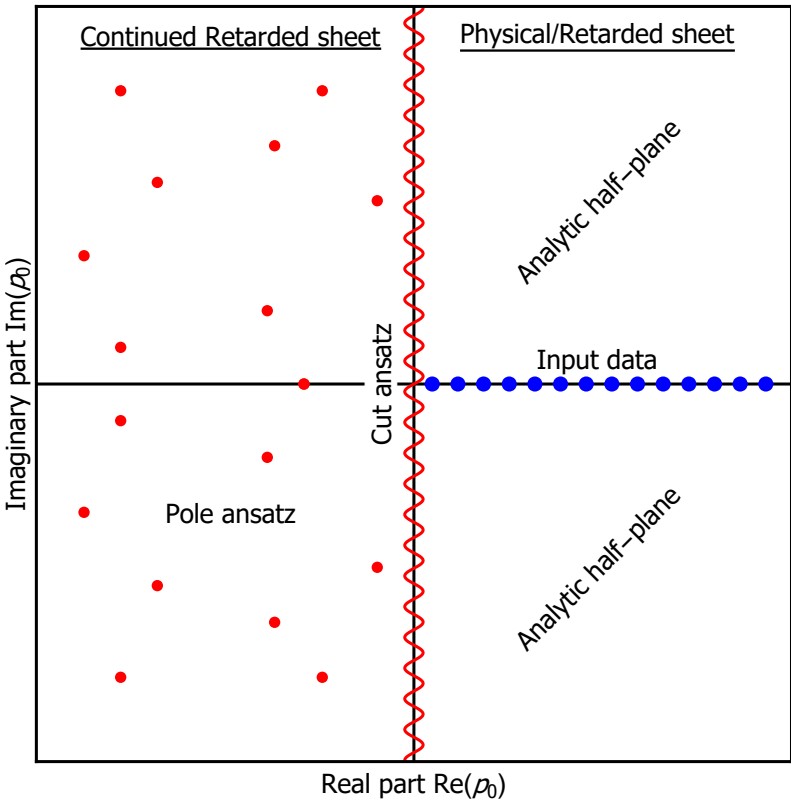

Figure 5: Schematic analytic structure of a retarded propagator. All non-analytic structures are in the $\mathrm{Re}\, p_0 < 0$ half-plane, reflecting the analyticity constraints from the existence of a spectral representation.

In the following we focus on the retarded propagator. However, all statements hold equivalently for the advanced propagator due to (33).

The finite imaginary part in the retarded propagator at $\mathrm{Re}\, p_0 = 0$ signals a branch cut and therefore a finite value of the spectral function, which is defined as the discontinuity of the propagator, c.f. (3). Being a holomorphic function for $\mathrm{Re}\, p_0 > 0$, the retarded correlation function can be analytically continued to the entire complex plane, where it is a meromorphic function since the propagator must vanish sufficiently fast for $p \to \infty$.

Our reconstruction approach is based on an ansatz for the complex structure of the analytically continued retarded propagator. This has the advantage that (1) holds trivially and it is possible to enforce (12) analytically. Furthermore, the branch cuts describing the IR and UV asymptotics can be implemented explicitly and in a straightforward manner.

The ansatz is build up from poles and polynomials. This is possible since the most important, i.e. physically relevant branch cuts, e.g. logarithms and square roots, can be constructed from a series of poles. Of course, branch cuts can also be taken into account directly. If one is only interested in the reconstruction of a spectral function itself, there is an additional freedom to choose the branch cut of e.g. logarithms, as long as they are in the meromorphic half-plane since it does not alter the result. Therefore a rather generic ansatz is the one depicted in Figure 5, where all cuts are chosen to be on the $\mathrm{Re}\, p_0 = 0$ axis.

## 5.2 Reconstruction method

Our approach is based on the ability to explicitly select an appropriate basis. As a direct consequence, prior knowledge about the spectral function, e.g. its asymptotics and its functional

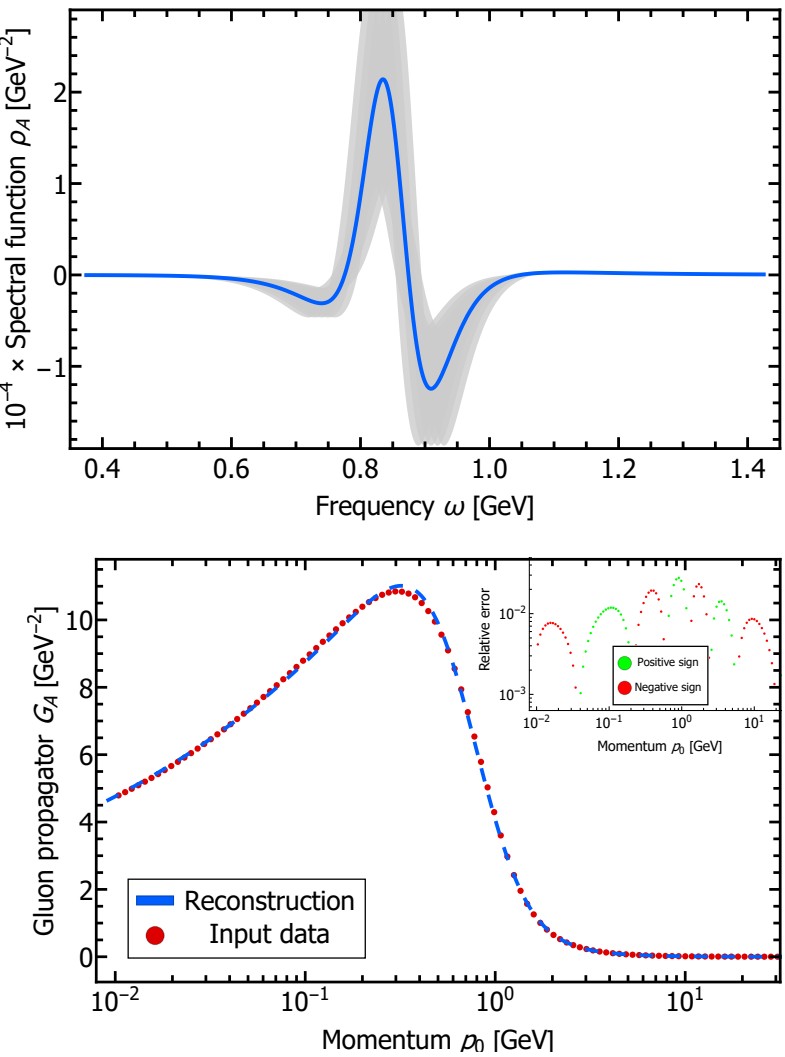

Figure 6: Top: Gluon spectral function. The solid blue line shows our best result. The gray band around it indicates our estimate for the systematic error. Bottom: Gluon propagator reconstructed from the spectral function shown in Figure 7 in comparison to the original propagator.

form in general, can and should be included into the basis. In turn, analytic calculations can serve as a guiding principle for choosing a suitable basis. Importantly, this does not fix the method used to determine the coefficients of the basis. Note that the functional bases deployed in most reconstruction procedures are chosen implicitly, such as e.g. in Bryan's MEM.

One might naively expect that by selecting a basis a priori the ill-conditioned problem of reconstructing the spectral function from Euclidean data becomes artificially regularized. In general this is not the case, as the number of different structures can be chosen arbitrarily large, as is also the case in most other reconstruction procedures. Our specific choice of basis only ensures that the asymptotic and analytic properties discussed above are met.

If the number of structures permitted by the basis function is larger than those actually encoded in the Euclidean correlator data, the problem remains ill-conditioned and Bayesian inference needs to be carried out, assigning a prior probability to the individual basis parameters. The state-of-the-art implementation of Bayesian inference, which provides insight into the full posterior probability distribution and not simply a maximum a posteriori estimate, rests

on Hamiltonian Monte Carlo (HMC) techniques (for the industry standard see MC-STAN [80]).

On the other hand one may systematically reduce the number of allowed structures in the basis ansatz until an ordinary $\chi^2$ fit becomes stable. If at the same time such a restricted basis still allows the Euclidean data to be reasonably well reproduced the corresponding basis parameters constitute a valid solution. This issue is discussed in a simple mock example in Appendix C.

### 5.3 Construction of a gluon propagator basis

We now introduce the explicit functional form of the basis used in the subsequent reconstruction of the spectral function. It consists of the several modular, dimensionless building blocks. We start with a set of generalized Breit-Wigner structures,

$$\hat{G}_{\text{Ans}}^{\text{pole}}(p_0) = \sum_{k=1}^{N_{\text{ps}}} \prod_{j=1}^{N_{\text{pp}}^{(k)}} \left( \frac{\hat{\mathcal{N}}_k}{(\hat{p}_0 + \hat{\Gamma}_{k,j})^2 + \hat{M}_{k,j}^2} \right)^{\delta_{k,j}} . \tag{34}$$

In addition, we introduce a polynomial-like structures

$$\hat{G}_{\text{Ans}}^{\text{poly}}(p_0) = \sum_{j=1}^{N_{\text{poly}}} \hat{a}_j \left( \hat{p}_0^2 \right)^{\frac{j}{2}} . \tag{35}$$

To capture the asymptotic IR and UV behaviors, we introduce the following factor,

$$\hat{G}_{\text{Ans}}^{\text{asy}}(p_0) = (\hat{p}_0^2)^{-1-2\alpha} \left[ \log\left( 1 + \frac{\hat{p}_0^2}{\hat{\lambda}^2} \right) \right]^{-1-\beta} . \tag{36}$$

The final ansatz is then given by the product of the three individual contributions (34), (35) and (36):

$$G_{\text{Ans}}(p_0) = \mathcal{K} \, \hat{G}_{\text{Ans}}^{\text{pole}}(p_0) \, \hat{G}_{\text{Ans}}^{\text{poly}}(p_0) \, \hat{G}_{\text{Ans}}^{\text{asy}}(p_0) , \tag{37}$$

where $\mathcal{K}$ only carries the appropriate dimension. The coefficients are constraint such that (1) holds analytically. The superconvergence (12) is not included analytically, however it is realized with high accuracy, we get back to this in Section 5.4.

### 5.4 Gluon spectral function reconstruction and benchmarking

With the explicit form of the basis laid out above, we can continue to extract the gluon spectral function from gluon propagator data obtained in the scaling scenario [18].

As a full HMC analysis of the gluon propagator data is beyond the scope of this paper. Instead, we choose the simpler strategy of systematically reducing the number of possible structures allowed by the ansatz. We arrive at a functional form, which permits us to reproduce the Euclidean data of the scaling scenario within $\sim 2\%$ relative deviation, as shown in the right panel of Figure 6. At the same time this restricted basis is simple enough that its parameters can be fixed by a standard $\chi^2$ fit. Our best fit uses $N_{\text{ps}} = 1$, $N_{\text{pp}}^{(1)} = 6$ and $N_{\text{poly}} = 5$ and leads to our final result, the gluon spectral function shown in the left panel of Figure 6. The shape of the result is stable against a small variation of $N_{\text{pp}}^{(1)}$ and $N_{\text{poly}}$. Nevertheless, we find degenerate solutions varying in their peak height, this is indicated by the grey band in the left panel of Figure 6.

The red dots in the right panel of Figure 6 denote the numerically evaluated input data from [18], while the dashed line represents the the Euclidean correlator corresponding to our

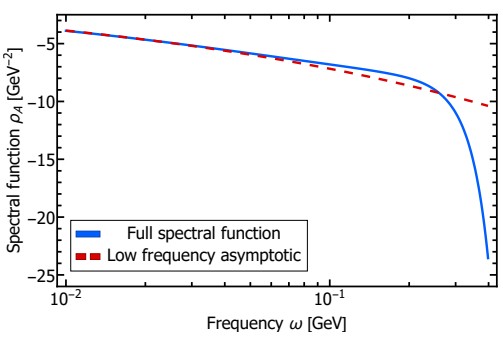 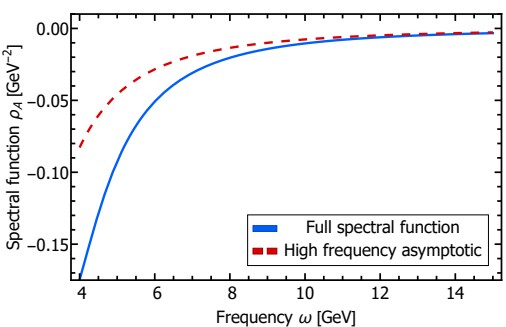

Figure 7: Low (left) and high (right) frequency behavior of our result for the spectral function shown in Figure 6. The dashed red lines show the asymptotic limits given by (23) and (16).

reconstructed spectral function. The inset shows the relative error on a logarithmic scale where deviations with positive sign are colored green, those with negative sign are colored red. While the coefficients $\alpha$ and $\beta$ in the asymptotic part of the basis functions are related to $\kappa$ and $\gamma$ of the IR and UV asymptotics, respectively, we note that they do not need to match exactly, since the former may be partially absorbed by some of the $\delta_{k,j}$'s. We list their values in Table 1 for completeness. The fit is heavily constraint, as we must enforce the complex structure as well as the correct asymptotics. As a consequence a useful and reliable error estimation is not possible.

Let us inspect the behavior of the reconstructed spectrum in more detail. From Figure 6 we infer that the fitted propagator, by construction, is able to reproduce the asymptotics of the UV and the IR very well. This directly translates into the correct asymptotic behavior of the spectral function in the IR and the UV, as shown in the left and right panel of Figure 7, respectively. The asymptotics are closely reproduced either below $\omega \approx 20\,\text{MeV}$ and above $\omega \approx 12\,\text{GeV}$.

Note that the well pronounced negative trough above the main positive peak in Figure 6 does not connect directly to the negative asymptotics but instead the spectrum returns into the positive once more before eventually becoming negative for good, i.e. approaching the frequency axis from below asymptotically. This behavior is reminiscent to what has been found in a previous lattice QCD study [6]. While the data there was not precise enough to capture the asymptotic behavior reliably, indications for a similar positive bump structure above a deep negative trough were found (see Figure 4).

Superconvergence (12) is not enforced analytically as it would unnecessarily complicate

Table 1: Parameters obtained in our best fit for the ansatz (37).

| $\hat{\mathcal{N}}_1$ | $\alpha$ | $\beta$ | $\hat{\lambda}$ | | |
|---|---|---|---|---|---|
| 1.33678 | -0.428714 | -0.777213 | 1.75049 | | |
| $\hat{a}_1$ | $\hat{a}_2$ | $\hat{a}_3$ | $\hat{a}_4$ | $\hat{a}_5$ | |
| 0.454024 | 0.241017 | 3.10257 | -1.30804 | 0.63701 | |
| $\hat{\Gamma}_{1,1}$ | $\hat{\Gamma}_{1,2}$ | $\hat{\Gamma}_{1,3}$ | $\hat{\Gamma}_{1,4}$ | $\hat{\Gamma}_{1,5}$ | $\hat{\Gamma}_{1,6}$ |
| 0.100169 | 0.100141 | 2.36445 | 1.5564 | 1.22013 | 1.15102 |
| $\hat{M}_{1,1}$ | $\hat{M}_{1,2}$ | $\hat{M}_{1,3}$ | $\hat{M}_{1,4}$ | $\hat{M}_{1,5}$ | $\hat{M}_{1,6}$ |
| 0.849883 | 0.849902 | 2.52171 | 2.44035 | 3.6016 | 2.36723 |
| $\delta_{1,1}$ | $\delta_{1,2}$ | $\delta_{1,3}$ | $\delta_{1,4}$ | $\delta_{1,5}$ | $\delta_{1,6}$ |
| 1.61116 | 1.94095 | -2.54586 | 1.89765 | 0.168592 | 0.296215 |

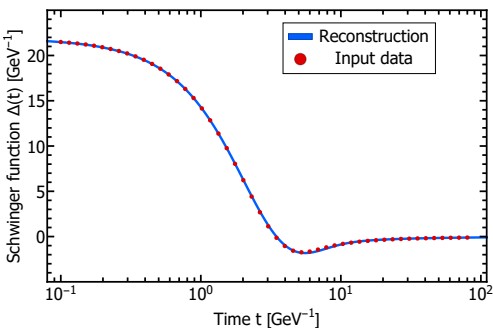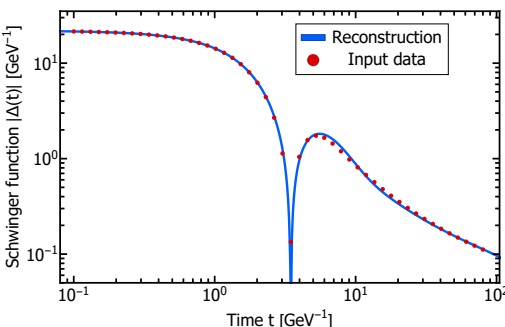

Figure 8: Schwinger function obtained from the reconstruction, blue line, and from the Euclidean input, red dots, in a semi-log (left) and log-log (right) depiction. See Figure 6 for the corresponding Euclidean propagators in momentum space.

our ansatz while being realized already very well on a numerical level as it is only violated by the branch point of the perturbative cut

$$\left(\int_0^\infty \mathrm{d}\eta \; |\eta\rho_{\mathrm{A}}(\eta)|\right)^{-1}\left(\int_0^\infty \mathrm{d}\eta \; \eta\rho_{\mathrm{A}}(\eta)\right) \approx 10^{-4}. \tag{38}$$

The height of the main positive and negative structure still show rather sizable uncertainties, which is related to our fit having been applied only to a precision of $\sim 2\%$.

The Schwinger function

$$\Delta(t) = 2\int_0^\infty \mathrm{d}p_0 \; e^{\mathrm{i}p_0 t} G(p_0) \tag{39}$$

is potentially more sensible to differences in the peak height of the spectral function as it corresponds to a Laplace transform of the spectral function, see e.g. [19]. The Schwinger functions from the reconstruction and the input data are shown in Figure 8. The point of the zero crossing between both result matches very well and the overall agreement is of the same level as for the Euclidean propagator. We interpret this as further evidence for our successful reconstruction of the gluon spectral function.

Performing a full Bayesian analysis, which allows for a robust reconstruction including more analytic structures in the basis, we expect the uncertainties of the reconstruction to reduce further. This is however postponed to future work.

# 6 Conclusion

We have discussed the reconstruction of the gluon spectral function in Landau gauge QCD from numerical Euclidean data, as well as its analytic properties. In particular, we have put forward a novel reconstruction approach, which possesses these analytic properties. It satisfies the Oehme-Zimmermann superconvergence relation, has the correct low and high frequency asymptotics, and reproduces the Euclidean gluon propagator data with $\sim 2\%$ accuracy. The key to this successful reconstruction lies in two novel ingredients:

The first one is the use of the analytic low frequency asymptotics of the gluon spectral function in the reconstruction. The latter is related to the IR asymptotics of the Euclidean propagator through the novel general relation (6) that has been derived in Section 2. The analytic knowledge of the spectral function for $\omega \to 0$ eliminates the typically large systematic

uncertainty in reconstruction methods at low frequencies, and hence may significantly improve the spectral reconstruction, independently of the used method.

The second novel ingredient in our approach originates in the careful analysis of the analytic structure of two-point correlation functions, and is described in Section 5. This analysis leads to an ansatz for the propagator in the complex plane that takes into account the generic pole and cut structure. The parameters of our quite general ansatz can then be determined from Euclidean data.

In our opinion these two novel ingredients will improve the accuracy of spectral reconstructions in general, and should be incorporated into existing Bayesian and non-Bayesian frameworks. This is briefly discussed in Appendix C.

We currently extent the present analysis to the finite temperature Euclidean data from [81], and the QCD correlation functions from [82]. This allows for an improved determination of transport coefficients following up on [1,4] as well as an access to hadronic observables.

## Acknowledgements

We thank M. Betancourt, D. Binosi, E. Grossi, A. Maas, D. Rosenblüh, A. Tripolt and F. Ziegler for discussions. This work is supported by the ExtreMe Matter Institute (EMMI) and the grant BMBF 05P12VHCTG. It is part of and supported by the DFG Collaborative Research Centre "SFB 1225 (ISOQUANT)".

## A  Details of the derivation of (6)

Our analysis that lead to the derivation of (6) in Section 2 is based on the existence of a spectral representation (1). The inverse relation (3) together with the existence of a spectral representation (1), that holds for the entire complex plane, has strong implications on the analytic structure of the propagator on the left hand side. As a consequence the holomorphicity of the associated retarded and advanced propagators in their respective domain of definition, c.f. Section 5.1, follows directly from Cauchy's theorem. As we define our spectral function as the cut in the propagator along the $\mathrm{Re}\, p_0$ axis, i.e. (3), the spectral representation is sufficient as condition. Having established analyticity in one half-plane, (6) holds already by means of the Cauchy-Riemann equations, where the additional factor two comes from (3).

In general neither the Minkowski propagator nor the spectral function is a function in the classical sense, but a tempered distribution. Nevertheless, the arguments about the analytic properties hold, see e.g. [83, 84]. Another prerequisite mentioned in the main text is the smoothness of the spectral function at zero. In the absence of truly distributional contribution to the integral for small $p_0$ in (1), the spectral function must go to zero sufficiently fast, otherwise the propagator obtains at least a log-divergence in $p_0$. On the other hand, if such contributions would be present this argument might not hold. A detailed discussion about the issue of these contributions regarding the gluon and their relation to functional methods can be found in [85, 86]. Mathematically rigorous statements in the context of axiomatic QFT's about this issues and the relation to our derivation are beyond the scope of this work.

# B  Poles of decoupling scenario I

More details about the poles of (26) are collected here We drop the normalization $Z_{\text{IR}}$ and additional notation to keep things simple

$$\hat{G}^{(\text{dec})}(p) = \left(\hat{m}_{\text{gap}}^2 + \hat{p}^2 \ln \hat{p}^2\right)^{-1}. \tag{40}$$

Using the Lambert W-Function, with the usual index notation for the different branches, the roots can be expressed as

$$z_{0,\pm}^{(r)} = \pm\sqrt{\psi_0}, \tag{41}$$

$$z_{-1,\pm}^{(r)} = \pm\sqrt{\psi_{-1}}, \tag{42}$$

with

$$\psi_k = e^{W_k(-\hat{m}_{\text{gap}}^2)}. \tag{43}$$

There are now three different cases for the additional poles

$\hat{m}_{\text{gap}}^2 < 1/e$

First order poles, located pairwise on the Euclidean axis $(\text{Im}(z_{k,\pm}^{(r)}) = 0)$.

$\hat{m}_{\text{gap}}^2 = 1/e$

Two second order poles at $p = \pm 1/\sqrt{e}$

$\hat{m}_{\text{gap}}^2 > 1/e$

Complex first order poles which are linked by complex conjugation (as required by Lorentz invariance)

$$z_{0,\pm}^{(r)} = \left(z_{-1,\pm}^{(r)}\right)^*. \tag{44}$$

Introducing the additional function

$$\chi_k = 1 + \ln \psi_k, \tag{45}$$

the residues can be computed in a straight forward manner

$$\text{Res}[\hat{G}^{(\text{dec})}, z_{k,\pm}^{(r)}] = (2\chi_k z_{k,\pm}^{(r)})^{-1} \tag{46}$$

for $k = -1, 0$. The additional term in (11) can then conveniently be written as

$$\text{pole term} = \sum_{k \in \{-1,0\}} \frac{1}{\chi_k \left(\hat{p}^2 - e^{W_k(-\hat{m}_{\text{gap}}^2)}\right)}. \tag{47}$$

# C  Other reconstruction approaches

In this appendix we discuss two reconstruction strategies, the Bayesian Reconstruction using a quadratic prior (Tikhonov) and the Schlessinger point method, which we had explored previous to developing the method presented in this paper. The reason that a new approach became necessary for the reconstruction of the gluon spectral function lies in the individual shortcomings of the above mentioned methods.

Our goal is to reconstruct a spectral function, which, as discussed in the main part of this paper, should exhibit the following properties:

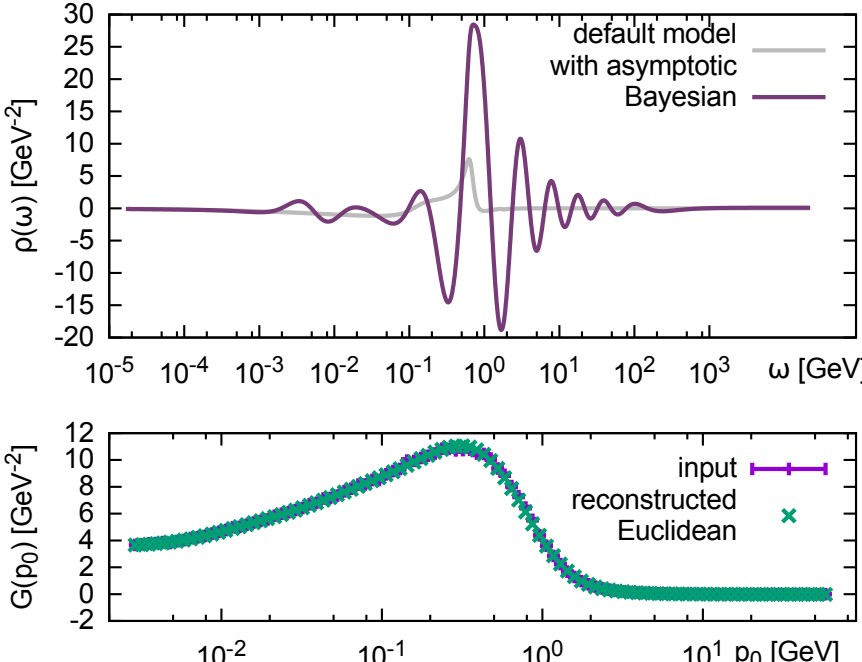

Figure 9: (top) Representative example of a Bayesian reconstruction of the gluon spectral function based on FRG Euclidean data in the scaling scenario. One hundred data points with a relative error of $10^{-2}$ were supplied, as well as a default model (gray solid), which produces the correct asymptotics of the correlator. We find that while the asymptotics are correctly recovered in the final result (solid violet) eventually, the agreement emerges far later than in our new method. In addition we find that the non-perturbative region at around $\omega \sim 1\,\text{GeV}$ is contaminated by strong ringing artifacts (bottom) Comparison between the reconstructed Euclidean correlator (green) and the input data (violet).

1. Normalization, as required by Oehme-Zimmermann superconvergence (12)

2. Correct low frequency asymptotics, c.f. Section 4

3. Correct high frequency asymptotics, c.f. Section 3

4. Respect the Källén–Lehmann spectral representation (1)

5. Antisymmetric around $\omega = 0$, c.f. (2)

6. A smooth function without drastic oscillations

A successful reconstruction based on Euclidean data should fulfill these requirement. We observed that neither the gBR method nor the Schlessinger point method was able to meet all the requirements in a satisfactory manner.

The underlying reason for these difficulties is related to the fact that both methods indirectly or directly choose a set of basis functions not adapted to the problem at hand.

### C.0.1 Bayesian reconstruction

The Bayesian reconstruction usually selects a set of basis functions simply by introducing a numerical integration scheme to represent the discretized spectral function. This choice of

basis is naturally unaware of the analytic structure of the correlator and thus of the functional form admissible for the spectral function. In particular this basis does not prevent highly oscillatory and thus unphysical structures to manifest themselves in the end result. In the spirit of the Bayesian approach the prior probability then needs to be constructed such that these oscillatory solutions are suppressed. The quadratic prior, we have found is not efficient in doing so and thus unphysical ringing may persists in the end result. Similar ringing also manifests itself in case of the generalized BR method.

We have performed reconstructions based on the Euclidean data in the scaling scenario, using different default models, which were endowed with the correct asymptotic form. One example is shown in Figure 9, where the green solid line denotes the default model and the solid violet line corresponds to the Bayesian end result. One hundred data points with a relative error of $10^{-2}$ were used. Due to the asymptotics supplied in the default model, as well as the fact that the Bayesian method uses the Källén–Lehmann representation to translate the spectrum into a Euclidean correlator, items (2-5) from our list are fulfilled here. Item (1) is not fulfilled since the ratio between the area under the curve and the area under the absolute value of the spectrum is around 0.8. The most striking drawback of this reconstruction however is the strong oscillatory behavior found, which renders an interpretation of the non-perturbative region at intermediate frequencies at best difficult. Note that with currently available lattice QCD data such a strong oscillatory behavior was not observed when investigating the gluon spectrum.

It will be interesting future work to include the prior information on the analytic structure of the gluon propagator into the prior probability of these Bayesian approaches.

### C.0.2 Backus-Gilbert reconstruction

The Backus-Gilbert approach to spectral function reconstruction operates [87, 88] with an implicit set of basis functions, which are characterized by the resolution function $\Delta(\omega - \omega')$. Also in this case the basis is not aware of the analytic structure admissible for the correlator under investigation. The naive Backus-Gilbert method in addition requires a regularization prescription, for which we here choose the Tickhonov approach with regulator parameter $\lambda$ [89].

No explicit default model enters the BG approach, i.e. the prior information needed to give meaning to the ill-conditioned inverse problem enters through the definition of the optimization functions of which the BG spectrum is an extrema. This functional is designed such that it selects a solution for which the resolution function $\Delta(\omega - \omega')$ is most sharply peaked.

Note that in order to carry out the BG reconstruction the so called response kernels need to be computed [89]. Here in case of the gluon spectrum these correspond to integrals over the Källén–Lehmann without the spectrum multiplied. As these functions are not well defined, we instead choose to reconstruct the function $\rho(\omega)\omega$ with the corresponding response Kernel

$$R(p_0) = \int_0^\infty \frac{d\omega}{\pi} \frac{1}{\omega^2 + p_0^2} \,, \tag{48}$$

which is finite. The spectrum is the obtained from dividing out $\omega$ from the raw reconstruction.

A series of results for the BG spectral reconstruction based on one-hundred ideal Euclidean input data points is shown in the top panel of Figure 10 for several different values of $\lambda$. We find that a main peak close to $\omega = 1$ GeV is consistently found and in addition the reconstructions show a negative trough close to the origin and above the main peak. However it is also clear from the top panel that close to the origin ringing artifacts again impede the physics interpretation of the result.

The BG solution by construction is not required to reproduce the input data, which then also leads to significant deviations as shown in the center panel of Figure 10. Thus, we find

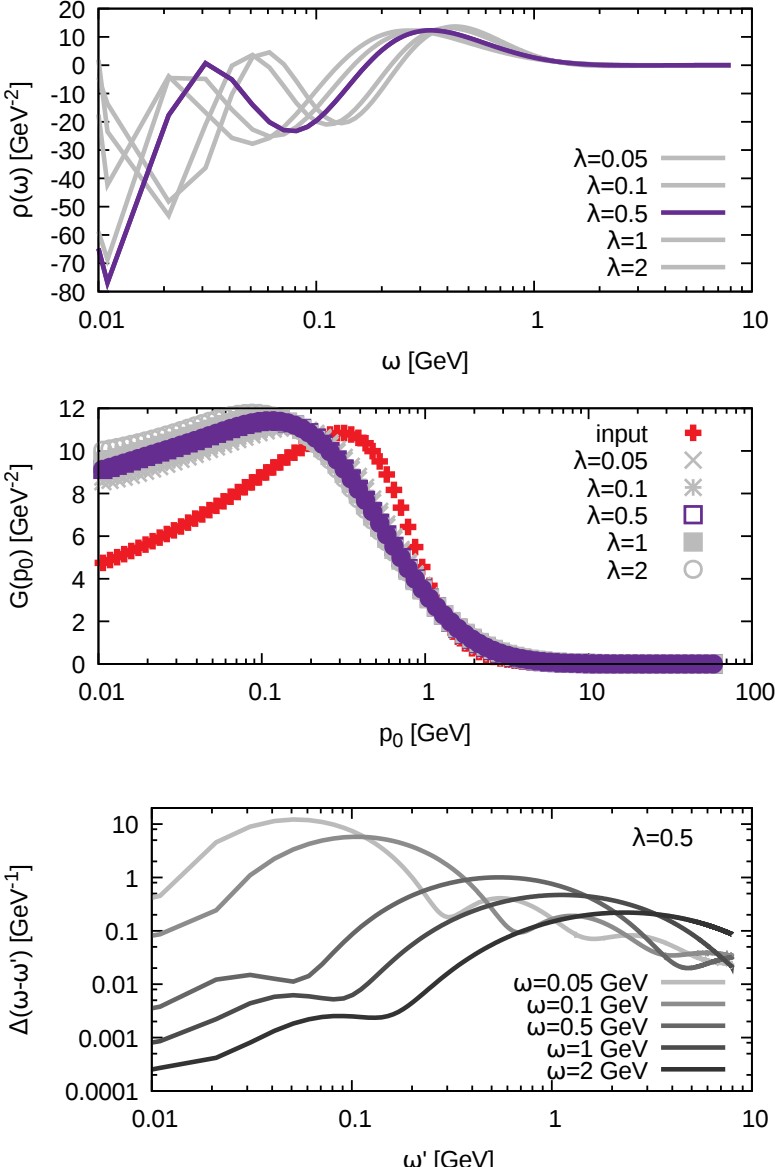

Figure 10: (top) Series of Backus-Gilbert reconstructed spectral functions using the Tickhonov regularization prescription for different regulator parameter $\lambda = [0.05..2]$. Note that the reconstruction identifies the presence of both the negative trough close to the origin and the one above the main peak at around $\omega > 1\,\text{GeV}$. Ringing at small frequencies is not cured by simply increasing the value of $\lambda$. The best choice $\lambda = 0.5$ is given in dark violet, while the other values are denoted by light gray curves. (center) Euclidean data of the reconstructed spectra compared to the original input (red). The result corresponding to $\lambda = 0.5$ (dark violet) works relatively well above $p_0 > 1\,\text{GeV}$ but misses the position of the main peak structure and exhibits too weak of a backbending. (bottom) Resolution function $\Delta(\omega - \omega')$ for the best choice $\lambda = 0.5$ plotted for completeness.

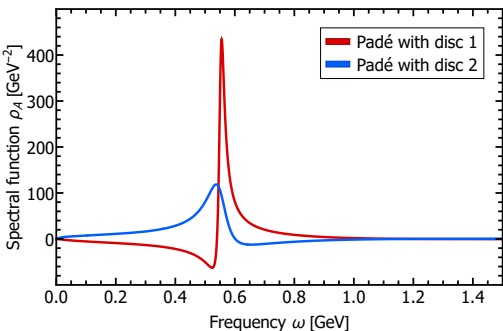 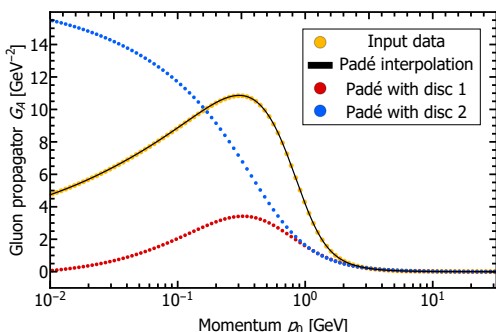

Figure 11: Reconstruction of the gluon propagator via Padé, the difference between the two spectral functions is described in the main text. The left panel shows the spectral functions while the right panel shows the input data, the Padé interpolant and the reconstructions obtained from the spectral functions.

that the BG approach in case of the gluon spectral function is challenged to meet, in particular, criteria (1-4) that we require for a successful reconstruction.

### C.0.3   Padé-type approaches

Padé approaches, e.g. Schlessinger's point method, obtain the spectral function from analytically continuing an interpolating or fitted rational function. The nomenclature *Padé* is used here in a loose sense, referring to all approaches based on matching a rational function of arbitrary degree to Euclidean data. Usually, (3) is used in order to obtain the spectral function. It has the advantage of being easy to apply and gives reasonable results for the position of the lowest lying resonances, see e.g. [90]. Padé approaches describe the analytic structure of the analytically continued retarded propagator in terms of poles and is therefore naturally contained in our approach outlined in Section 5.

However, it does not respect the holomorphicity of the retarded propagator in the given half-plane, c.f. Figure 5 and the corresponding discussion, by design. While Padé approaches will still converge for infinite precision and infinitely many data points, violations of the holomorphicity at finite precision might be severe and make an unambiguous reconstruction impossible. As the spectral function obtained from (3) does not respect (1) anymore, if the holomorphicity of the right half-plane is violated, the obtained spectral function will fail to reproduce the Euclidean propagator. The violation might be acceptable if the violation is small, suitably defined via the reconstruction. But is most certainly not, if a pole dominantly contributing in (1) is missing. A more realistic spectral function can then still be obtained by suitably modifying (3), but there is no consistent, unambiguous way of doing so.

This discussion is especially relevant when turning to the reconstruction of the gluon, since here the dominant pole violates the holomorphicity of the retarded propagator. We employed two choices for the reconstruction, "Padé with disc 2", where we keep (3) with a flipped sign to account for the dominant pole being in the wrong half-plane, and "Padé with disc 1", but evaluated it at an argument with a finite real part slightly larger than the position of the pole. The corresponding spectral functions are shown in the left panel of Figure 11. Both methods get the roughly a similar shape which is also loosely compatibly with our main result Figure 6. However, "Padé with disc 2" fails to reproduce the general shape in the reconstruction and "Padé with disc 1" gets a significantly too small propagator, but describes the shape correctly. Additionally several other requirements listed above are not fulfilled. While some requirements might be fixed by manipulating poles in the Padé interpolant, any systematic way of doing so will lead to an approach very similar to the one described in the main text in

Section 5.

## C.1 Mock reconstruction benchmark

In this section we demonstrate with the simple example of a two Breit-Wigner spectral function how incorporating our novel approach into a Bayesian framework allows to straight forwardly improve the spectral reconstruction. As mock spectral function we take the parametrization of (10) with a direct sum of two Breit-Wigner peaks

$$\rho_{\text{mock}}(\omega) = \rho_1^{\text{(BW)}}(\omega) + \rho_2^{\text{(BW)}}(\omega),$$ (49)

and the following values for peak positions and widths

| Parameter | $A$ | $M$ | $\Gamma$ |
|---|---|---|---|
| Peak 1 | 0.35 | 1.0 | 0.25 |
| Peak 2 | 0.65 | 3.0 | 0.25 |

From this spectrum we evaluate 60 equidistantly spaced Euclidean correlator data points in the imaginary frequency interval between $0 - 45$ GeV, which are subsequently salted with Gaussian noise with a strength leading to $10^{-3}$ relative errors.

In the absence of any prior knowledge about the analytic structure (i.e. the two poles) contained in our example data we may choose to deploy a standard Bayesian method, such as the BR method [91], which only enforces positive definiteness and smoothness. As shown in Figure 12 as green solid line, with the provided quality of the Euclidean data, this method manages to correctly identify the number of peak present but only achieves an accuracy of the peak positions of around $75 - 80\%$.

Now we can proceed to deploy our novel ansatz for the functional basis in (37). Three different cases are possible. Depending on the number $N_{\text{ps}}$ of pole structures chosen in the basis, we may either have less structure, exactly the same amount of pole structures, or irrelevant additional structure present compared to what is actually encoded in the Euclidean data.

In the first case, where only one pole is contained in the basis, fitting its parameters is a well posed problem and can be carried out using both a naive $\chi^2$ fit or a full Bayesian analysis, where each parameter is endowed with an additional prior probability. In Figure 12 we have carried out a full Bayesian estimation of the posterior for the parameters in the single pole basis (purple dashed line) using the Hamiltonian Monte Carlo framework implemented by the MC-STAN [80, 92] library. The used prior only enforces the finiteness of the peak position. As expected this too strongly restricted basis yields a reconstruction, which cannot correctly account for the pole structure and instead positions one peak in between the actual two peaks present.

Increasing the number of available poles in the basis to two, we have the same number of structures in our basis, as we have in our data. This case is still well-posed and again admits a solution both via $\chi^2$ fit and a full Bayesian analysis. Since we are using a parametrization that can exactly match our input data it is not surprising that now the end result from the HMC analysis lies spot on the mock spectrum (orange dashed curve). Note that for the $\chi^2$ fit the result is slightly worse and only reproduces the mock spectrum with around 5% deviation.

Please note that as soon as two or more poles are present in the basis an ordering prescription must be enforced, otherwise the posterior distribution degenerates by means of a simple reparameterization.

The more interesting case, relevant in practice concerns allowing more structure in the basis than encoded in the Euclidean data, which we here tests with a three pole basis ansatz. In view of the $\chi^2$ fit this problem is now ill-conditioned and indeed carrying out a naive fit shows

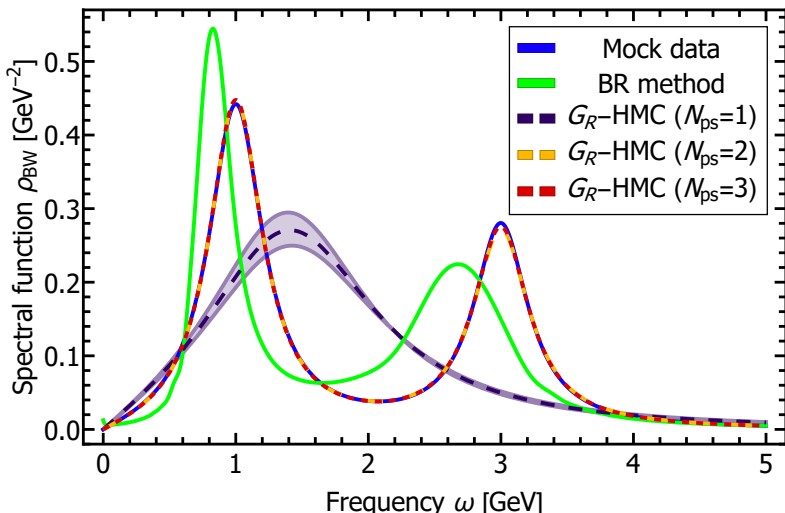

Figure 12: Reconstruction benchmark with a double Breit-Wigner peak.

that the obtained parameters become unstable. Here the HMC approach plays out its strength, maintaining stability even if the problem becomes ill-conditioned. Indeed the position and width of the lowest two pole structures is again recovered excellently (red dashed curve), while the posterior of the position of the third peak shows that it is highly unconstrained and thus irrelevant. The pole ordering naturally achieves that any excess pole structure beyond what is encoded in the data is simply pushed to infinity, not contributing to the end result.

We find that incorporating our novel basis may significantly improve the reconstruction result compared to those methods, which do not make any assumptions on the analytic structure of the underlying data. The example used here of course is simplistic but captures a main aspect, a multi pole structure, encountered also in more realistic cases.

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
