# Peer review of "Reconstructing the gluon"

_SciPost Physics, doi:SciPost Phys. 5, 065 (2018)_

## Round 2 · Referee Report · Anonymous · 2018-9-10

Strengths

1-A general relation between the behaviour of the bosonic spectral function at low frequencies and the Euclidean correlator in the infra-red has been derived.
2-A critical discussion of possible issues with the reconstruction of the gluon spectral function is given.
3-Illustrative examples are given to highlight the strengths of the new method which is a combined lattice-functional approach with improved analytic methods.

Weaknesses

1-Nothing worth to be mentioned.

Report

The paper “Reconstructing the gluon” discusses the reconstruction of the gluon spectral function in a special gauge from numerical (lattice) data for the gluon propagator. In order to do so, first the asymptotic behaviour of the spectral function at low frequencies is derived. Then, based on the analytic properties of the propagator, a suitable functional basis is used which allows for the reconstruction procedure.

It is somewhat surprising that such a general relation between the behaviour of the bosonic spectral function at low frequencies and the Euclidean correlator in the infra-red has not been recognized before. Nevertheless, the derivation of (6) is convincing and the given example is illustrative.

To apply (6) requires, of course, a sufficiently well established knowledge about the infra-red Euclidean propagator. For the latter different scenarios are discussed. In particular, sources for the $p^2\ln p^2$ behaviour in the decoupling solution are highlighted and it is made clear that while the general form might be well motivated, details are less clear e.g. because of limitations in the Euclidean data. Since with currently available lattice data this is not possible, the authors advertise a combined lattice-functional approach with improved analytic methods. I very much appreciate the critical discussion of the subject. Eventually (only) existing numerical data for the gluon propagator in the scaling scenario are used to reconstruct the gluon spectral function.

For the reconstruction, an ansatz for the retarded propagator in the entire complex plane is made, an appropriate basis is selected such that the asymptotic behaviour and the general functional form of $\rho$ is respected, and unknown parameters are determined within the analysis. The advantage of the proposed method becomes clear from Fig. 9 and is explained by a mock example in an Appendix which highlights that prior knowledge of the underlying analytic structures (including the criteria mentioned in Appendix C) is actually crucial for the reconstruction.

This is an excellent and exciting study. The reference list is extensive and gives a fair account of the existing literature. I look forward to the future studies mentioned as an outlook. I recommend publication in SciPost Physics but propose the following optional things to take into consideration:
1) It would be nice if the authors could leave a sentence or two about the physical consequences of the “positivity violation” in the spectral function.
2) The meaning of $\kappa$ could already be explained together with (17) and (18).
3) Maybe this is sufficiently discussed already but how general is the structure of the ansatz in (36). Could one imagine more?
4) How are the error estimates in Fig. 6 (left) obtained? How are the systematics estimated? To what extent does this contain the fact that the actual analysis was done with a reduced set of allowed structures in (36)?
5) What remains somewhat unclear: Does it turn out that the logarithmic corrections (as summarized on p.7) are crucially important for the reconstruction? Does this have consequences for the “positivity violation”?
6) Minor things: Below (10) probably “low frequencies” is meant. Below (12) “negative” is missing. Before (17) a reference might be handy.

Requested changes

1-Nothing but taking the optional changes into consideration.

Attachment

  • validity: high
  • significance: high
  • originality: high
  • clarity: high
  • formatting: excellent
  • grammar: excellent

Author:  Nicolas Wink  on 2018-11-07  [id 337]

(in reply to Report 1 on 2018-09-10)
Category:
remark
answer to question

Comment 1):
It would be nice if the authors could leave a sentence or two about the physical consequences of the “positivity violation” in the spectral function.
Answer:
We have extended the note on positivity violation at the end of Section III.

Comment 2):
The meaning of $\kappa$ could already be explained together with (17) and (18).
Answer:
We have shortly introduced $\kappa$ between (17) and (18), but left any further discussion in place since it is better suited at the individual IR scenarios.

Comment 3):
Maybe this is sufficiently discussed already but how general is the structure of the ansatz in (36). Could one imagine more?
Answer:
As mentioned in the text, the structures are chosen with a physical interpretation in mind. For different applications one can imagine more, and also requires it, e.g. different cut structures proportional to occupation numbers at finite temperature. For our text itself, we think this is sufficiently discussed in Section V A), Section V B) and Appendix A.

Comment 4):
How are the error estimates in Fig. 6 (left) obtained? How are the systematics estimated? To what extent does this contain the fact that the actual analysis was done with a reduced set of allowed structures in (36)?
Answer:
The error is estimated by degenerate solutions we find within the $\chi^2$ fit and systematics errors are, inherently by design, not estimable in such a problem. We find stability against a slight variation in the number of basis elements, which is a necessary condition for our reconstruction, but allowing for arbitrary structures in the Ansatz reintroduces the ill-conditioning of the problem. Choosing the basis elements with physical/analytic motivations is absolutely necessary. We have addressed these issues at the end of the second paragraph in Section V D).

Comment 5):
What remains somewhat unclear: Does it turn out that the logarithmic corrections (as summarized on p.7) are crucially important for the reconstruction? Does this have consequences for the “positivity violation”?
Answer:
As outlined in Section III, positivity violation is already present in the UV and therefore independent of the IR behaviour. Nevertheless, the logarithmic corrections do not spoil the positivity violation in the IR, this is outlined in Section IV C). For practical reconstructions, this should be reflected in an appropriate Ansatz, i.e. the logarithmic IR corrections should be present in (35). However, since we only considered the reconstruction within the scaling scenario we cannot make a statement about their practical importance. This will be considered elsewhere.

Comment 6):
Minor things: Below (10) probably “low frequencies” is meant. Below (12) “negative” is missing. Before (17) a reference might be handy.
Answer:
Indeed "low frequencies" was meant and "negative" was missing. Unfortunately there is not a single reference for (17), since it is based on a collection of references, which are listed for the scaling scenario below (20). And for the decoupling scenario we have an extensive discussion in Section IV B) (including references) that motivate (17). We have added a comment below (18) that indicates this.

---

## Round 2 · Referee Report · Anonymous · 2018-9-19

Strengths

1- This paper provides a very good in-depth discussion of the issues involved in reconstructing the spectral function from the Euclidean propagator. The more technical numerical points discussed in the appendix are particularly informative. $\\$

2- The reference list is extensive and provides important context for the issues discussed throughout the paper. $\\$

3- The paper attempts to develop a genuinely new reconstruction approach using a mixture of analytic and numerical techniques.

Weaknesses

1- The main weakness is that there appears to be issues with the central analytic result of the paper [Eq. (6)], or at least further clarification is required.

Report

In this work the authors propose a novel approach for investigating the structure of the Landau gauge gluon spectral function. By imposing a series of analytic consistency criteria, the authors select an appropriate functional basis which allows them to reconstruct the form of the spectral function given numerical data of the Euclidean propagator. $\\$

This is an interesting study, and it's particularly nice to see that the authors are attempting to develop a different numerical approach for extracting information about the spectral content of the gluon. However, there are a few issues which I feel require further clarification, in particular with regards to the analytic relation derived in Eq. (6) of the paper. $\\$

In Eq. (1) the authors write down the Euclidean spectral representation of the propagator. The representation they write down is correct except that the spectral function must depend on $\lambda^{2}$ and not just $\lambda$. In other words, the representation has the form: $G(p_{0}^{2}) = \int_{0}^{\infty} \frac{d\lambda}{\pi}\frac{\lambda\rho(\lambda^{2})}{\lambda^{2}+p_{0}^{2}}$. Since $\rho(\lambda^{2})$ is non-vanishing only for $\lambda^{2} \in [0, \infty)$, Eq. (2) is rather confusing since the spectral function itself is not defined for negative values of $\lambda^{2}$. What exactly is meant by the spectral function being anti-symmetric? If one ignores this issue and takes the derivative with respect to $p_{0}$, it is argued that one has the relation
\begin{align*}
\partial_{p_{0}}G(p_{0}^{2}) = -\int_{-\infty}^{\infty} \frac{d\lambda}{\pi}\lambda p_{0}\frac{\rho(\lambda^{2})}{(\lambda^{2}+p_{0}^{2})^{2}}
\end{align*}
which after taking the limit $p_{0} \rightarrow 0^{+}$ on both sides, and using the Poisson kernel representation of the Dirac delta, gives
\begin{align*}
\lim_{p_{0} \rightarrow 0^{+}}\partial_{p_{0}}G(p_{0}^{2}) = \frac{1}{2}\int_{-\infty}^{\infty} d\lambda \, \frac{d\delta(\lambda)}{d\lambda} \rho(\lambda^{2}) = -\frac{1}{2}\int_{-\infty}^{\infty} d\lambda \, \delta(\lambda) \frac{d \rho(\lambda^{2})}{d\lambda} = -\frac{1}{2}\lim_{\omega \rightarrow 0^{+}}\partial_{\omega}\rho(\omega^{2})
\end{align*}
Now the dependence of $\rho$ on $\omega^{2}$ becomes important because the derivative is with respect to $\omega$ and not $\omega^{2}$. By changing variables to $s=\omega^{2}$, the above relation takes the form
\begin{align*}
\lim_{p_{0} \rightarrow 0^{+}}\partial_{p_{0}}G(p_{0}^{2}) = -\lim_{s \rightarrow 0^{+}} \sqrt{s} \partial_{s} \rho(s)
\end{align*}
Assuming that $\partial_{s} \rho(s)$ is well-behaved at $s=0$, it follows that $\lim_{s \rightarrow 0^{+}} \sqrt{s} \partial_{s} \rho(s)=0$, and therefore Eq. (6) no longer appears to provide a non-trivial connection between the asymptotic behaviour of the propagator and its corresponding spectral function. It may well be that the statements made above are invalidated under certain conditions on $\rho(s)$, and a non-trivial connection between the low-frequency spectral function and infra-red propagator does indeed exist, in which case the authors need to specify precisely what these conditions are, and whether they have any bearing on their results. $\\$

Here are some more general comments and optional points to take into consideration:
$\\$
(1) In Eq. (11) it is assumed that the spectral function has a sequence of (complex) poles and a continuous contribution $\rho_{A}(\lambda)$. Is $\rho_{A}(\lambda)$ completely arbitrary, or are there certain conditions imposed on this component? For example, does $\rho_{A}(\lambda)$ vanish at $\lambda^{2} = M_{j}^{2}$? There is also a typo in Eq. (11): the LHS should presumably read $G_{A}(p_{0})$, not $G_{A}(\omega)$. $\\$

(2) It is stated that Eq. (20) follows from Eq. (6). As with Eq. (6), it would be appreciated if the authors could provide at least a sketch of this argument. $\\$

(3) In Fig. 6 the systematic error on the gluon spectral function is given. How exactly is this estimated? Also, given this error on the spectral function, is it possible to translate this into an error band on the reconstructed propagator? It would perhaps be interesting to see how this uncertainty translates. $\\$

(4) The authors plot the Schwinger function $\Delta(t)$ in Fig 8. As another check of superconvergence the authors might also consider plotting $\dot{\Delta}(t)$, which is proportional to the integral of the spectral function at $t=0$ (see e.g. the discussion in section III of 1310.7897). $\\$

In summary, I find this study interesting but I cannot recommend publication in SciPost Physics until the specific points I've raised, in particular with regards to Eq. (6), have been addressed.

Requested changes

1- Address the issues raised in the report with regards to Eq. (6). The other suggestions are optional.

Attachment

  • validity: ok
  • significance: good
  • originality: high
  • clarity: high
  • formatting: excellent
  • grammar: excellent

Author:  Nicolas Wink  on 2018-11-08  [id 338]

(in reply to Report 2 on 2018-09-19)
Category:
remark
answer to question
objection

Major issues:

Comment:
The major criticism regarding equation (6)
Answer:
We are aware that the spectral function is fully determined by positive frequencies, but whether or not the argument of the spectral function $\rho$ is linear,
i.e. $\rho(\lambda)$ or quadratic $\rho(\lambda^2)$ is a matter of convention
and must be dealt with consistently. From the viewpoint of analytic continuation and the differentiation between the advanced, retarded and Euclidean
propagator, the linear argument is very natural and therefore preferred. For example, Fig 5 is very simple when considered with a linear argument, but rather
complicated and counterintuitive when considered with a squared argument. Therefore we used a linear argument everywhere, with the only exception of
some parts of the IR scenarios since there the squared argument is more natural, but nevertheless we did so consistently. The different conventions come
with some very simple consequences, for the linear argument $\rho(\lambda)$ the spectral function is defined for $\lambda\in(-\infty,\infty)$ and is antisymmetric $\rho(\lambda) = -\rho(-\lambda)$,
while for the squared argument $\rho(\lambda)$ the spectral function is only defined for $\lambda\in[0,\infty)$.

Minor issues:

Comment:
In Eq. (11) it is assumed that the spectral function has a sequence of (complex) poles and a continuous contribution $\rho_A(\lambda)$. Is $\rho_A(\lambda)$ completely arbitrary,
or are there certain conditions imposed on this component? For example, does $\rho_A(\lambda)$ vanish at $\lambda^2=M_j^2$?
Answer:
$\rho_A(\lambda)$ is restricted in the sense that it must should be a valid spectral function, i.e. belong to a certain class of real valued distributions, which
is further elaborated it Appendix A. The additional complex conjugated poles, at position $M_j$, do not harm our study and are sometimes considered in the
literature and therefore included.

The requirement $\mathrm{Im} M_j \neq 0$ removes any potential ambiguity between $\rho_A(\lambda)$ and the additional complex conjugate poles and
therefore no further conditions are necessary.

Comment:
There is also a typo in Eq. (11): the LHS should presumably read $G_A(p_0)$, not $G_A(\omega)$.
Answer:
We have corrected the typo.

Comment:
It is stated that Eq. (20) follows from Eq. (6). As with Eq. (6), it would be appreciated if the authors could provide at least a sketch of this argument.
Answer:
We have added an extension of the explanation of the derivation of (20) below (20) to clarify the relation between the sign of $\rho_A(\omega)$ and
$\rho_A'(\omega)$. Please keep in mind that all statements about the limiting behaviour are understood in an asymptotic sense, i.e. they do not only
contain the value of the limit itself, but also its asymptotic behaviour.

Comment:
In Fig. 6 the systematic error on the gluon spectral function is given. How exactly is this estimated? Also, given this error on the spectral function,
is it possible to translate this into an error band on the reconstructed propagator? It would perhaps be interesting to see how
this uncertainty translates.
Answer:
The error is estimated by degenerate solutions we find within the $\chi^2$ fit. We find stability against a slight variation in the number of basis elements,
which is a necessary condition for our reconstruction, but allowing for arbitrary structures in the Ansatz reintroduces the ill-conditioning of the problem.
Choosing the basis elements with physical/analytic motivations is absolutely necessary.
We have addressed these issues at the end of the second paragraph in Section V D).

The different propagators obtained from other solutions are numerically equivalent to the shown reconstructed propagator (relative deviation on the level of
$10^{-3}$). A more detailed analysis with the outlined HMC method in order to obtain more reliable errors of the statistical reconstruction will be the content
of future work.

Comment:
The authors plot the Schwinger function $\Delta(t)$ in Fig 8. As another check of superconvergence the authors might also consider plotting $\dot\Delta(t)$,
which is proportional to the integral of the spectral function at t = 0 (see e.g. the discussion in section III of 1310.7897).
Answer:
We have already investigated and discussed the issue of superconvergence and find very good agreement as it is practically implemented in an analytically
manner within our approach.

---

## Round 3 · Author Response

We have included several suggestions from the first two referee reports.

---

## Round 3 · List of Changes

Main changes: - Extended the reasoning behind the error band in Fig. 6 - Added a note about the consequences of positivity violation - More details on the derivation of (20) - Introduced $\kappa$ between (17) and (18)

Other changes: - Corrected several typos - Different hyperlink styling for appendices

---

## Editorial Decision

published